# Fast Lifelong Adaptive Inverse Reinforcement Learning from Demonstrations

**Letian Chen\*, Sravan Jayanthi\*, Rohan Paleja**
**Daniel Martin, Viacheslav Zakharov, Matthew Gombolay**
Georgia Institute of Technology
Atlanta, GA 30332
{letian.chen, sjayanthi, rpaleja3, dmartin1, vzakharov3,
matthew.gombolay}@gatech.edu

**Abstract:** Learning from Demonstration (LfD) approaches empower end-users to teach robots novel tasks via demonstrations of the desired behaviors, democratizing access to robotics. However, current LfD frameworks are not capable of fast adaptation to heterogeneous human demonstrations nor the large-scale deployment in ubiquitous robotics applications. In this paper, we propose a novel LfD framework, Fast Lifelong Adaptive Inverse Reinforcement learning (FLAIR). Our approach (1) leverages learned strategies to construct policy mixtures for fast adaptation to new demonstrations, allowing for quick end-user personalization, (2) distills common knowledge across demonstrations, achieving accurate task inference; and (3) expands its model only when needed in lifelong deployments, maintaining a concise set of prototypical strategies that can approximate all behaviors via policy mixtures. We empirically validate that FLAIR achieves *adaptability* (i.e., the robot adapts to heterogeneous, user-specific task preferences), *efficiency* (i.e., the robot achieves sample-efficient adaptation), and *scalability* (i.e., the model grows sublinearly with the number of demonstrations while maintaining high performance). FLAIR surpasses benchmarks across three control tasks with an average 57% improvement in policy returns and an average 78% fewer episodes required for demonstration modeling using policy mixtures. Finally, we demonstrate the success of FLAIR in a table tennis task and find users rate FLAIR as having higher task ($p < .05$) and personalization ($p < .05$) performance.

**Keywords:** Personalized Learning, Learning from Heterogeneous Demonstration, Inverse Reinforcement Learning

## 1 Introduction

Robots are becoming increasingly ubiquitous with recent advancements in Artificial Intelligence (AI), largely due to the success of Deep Reinforcement Learning (DRL) techniques in generating high-performance continuous control behaviors [1, 2, 3, 4, 5, 6, 7, 8]. However, DRL's success heavily relies on sophisticated reward functions designed for each task. These hand-crafted reward functions typically require iterations of fine-tuning and consultation with domain experts to be effective [9]. Instead, Learning from Demonstration (LfD) approaches democratize access to robotics by having users demonstrate the desired behavior to the robot [10], removing the need for per-task reward engineering. While LfD research strives to empower end-users with the ability to program novel behaviors onto robots, we must consider that end-users may adopt varying preferences and strategies in how they complete the same task [11]. An LfD framework that assumes homogeneity across the set of provided demonstrations could cause the robot to fail to infer the accurate intention, resulting in unwanted or even unsafe behavior [12, 13]. On the other hand, embracing individual preferences can help robots achieve better performance and long-term acceptance from humans [14].

While personalization is important for accurate recovery of the demonstrator's behavior, personalization can also prove inefficient if each individual policy must be inferred separately. To avoid

---

* denotes equal contribution

6th Conference on Robot Learning (CoRL 2022), Auckland, New Zealand.

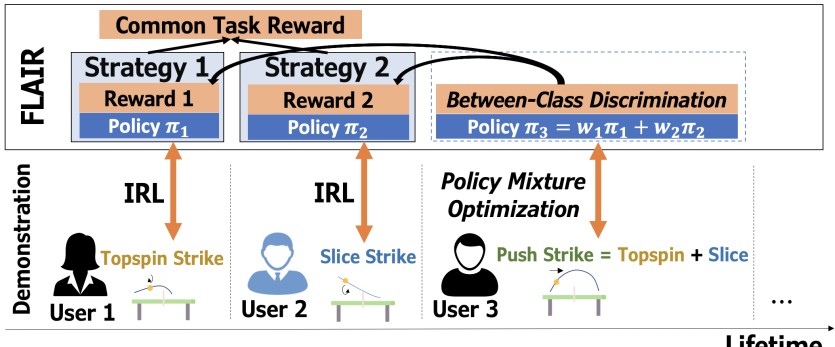

Figure 1: This figure shows an illustration of the lifelong learning process with our proposed method, FLAIR. As each demonstrator performs their strike, FLAIR determines whether the demonstration is novel. If a demonstration can be explained by a *policy mixture* of previously learned strategies, FLAIR accepts the policy mixture without training a new strategy. If the policy mixture is not close to the demonstration, FLAIR creates a new strategy and a prototype policy for the demonstration.

this issue, prior work, MSRD [15], decomposed shared and individual-specific reward information across heterogeneous demonstrations (i.e., demonstrations seeking to accomplish the same task with different styles). While MSRD significantly improves the accuracy and efficiency in personalized policy modeling, the framework must be trained all-at-once and is unable to handle *incremental/lifelong learning*, a more realistic paradigm for LfD real-world applications.

In this work, we develop FLAIR: Fast Lifelong Adaptive Inverse Reinforcement learning. As a running example, consider a series of humans teaching a robot how to play table tennis, a compelling robot learning platform [16, 17, 18]. Users of the robot may have their own preferences for table tennis strikes. As shown in Figure 1, the first user demonstrates a topspin strike, while the second user demonstrates a slice strike. The third user demonstrates a push strike, which could be explained by a composition of known behaviors of the previously seen topspin and slice prototypical behaviors.

Unlike prior LfD algorithms, FLAIR is capable of continually learning and refining a set of prototypical strategies either to (1) efficiently model new demonstrations as mixtures of the acquired prototypes (e.g., the third user in our example) or (2) incorporate a new strategy as a prototype if the strategy is sufficiently unique (e.g., the second user). Consider a real-world example where household robots are delivered to users' homes and the users want to teach those robots skills over the course of the deployment. User demonstrations from different end-users form a demonstration sequence the robots personalize to. In such a lifelong learning scenario, FLAIR autonomously identifies prototypical strategies, distills common knowledge across strategies, and precisely models each demonstration as prototypical strategies or policy mixtures. We show FLAIR accomplishes *adaptivity*, *efficiency*, and *scalability* in LfD tasks in simulated and real robot experiments:

1. **Adaptive Learning**: We display the *adaptivity* of FLAIR by successfully personalizing to heterogeneous demonstrations on three simulated continuous control tasks. FLAIR models demonstrations better than best benchmarks and achieves an average of 57% higher returns on the task.
2. **Efficient Adaptation**: FLAIR is more *efficient*, empirically needing an average of 78% fewer samples to model demonstrations compared to training a new policy.
3. **Lifelong Scalability**: We showcase the *scalability* of FLAIR in a simulated experiment obtaining 100 demonstrations sequentially. FLAIR identifies on average eleven strategies and utilizes *policy mixtures* to achieve a precise representation of each demonstration, providing empirical evidence for FLAIR's ability to learn a compact set of prototypical strategies in lifelong learning.
4. **Robot Demonstration**: We demonstrate FLAIR's ability to successfully leverage *policy mixtures* to achieve stronger task and personalization performance than learning from scratch in a real-world table tennis robot experiment.

## 2   Related Work

Two common approaches in LfD are to either directly learn a policy, i.e., Imitation Learning (IL), or infer a reward to train a policy, i.e., Inverse Reinforcement Learning (IRL) [19]. IL learns a direct

mapping from states to the actions demonstrated [20, 21]. Although a straightforward approach, IL suffers from correspondence matching issues and is not robust to changes in environment dynamics due to its mimicry of the demonstrated behaviors [22, 23]. IRL, on the other hand, infers the demonstrator's latent intent in a more robust and transferable form of a reward function [24].

Although traditional IRL approaches often overlook heterogeneity within demonstrations, there has been recent work that models heterogeneous demonstrations [25, 26, 27, 28, 29, 30]. One intuitive way is to classify demonstrations into homogeneous clusters before applying IRL [11]. The Expectation-Maximization (EM) algorithm also operates on a similar idea and iterates between E-step and M-step, where E-step clusters demonstrations and M-step solves the IRL problem on each cluster [31, 32]. When the number of strategies is unknown, a Dirichlet Process prior [33, 34, 35] or non-parametric methods [36] could be used. In these approaches, each reward function only learns from a portion of the demonstrations, making them prone to the issue of reward ambiguity [15]. Furthermore, these methods assume access to all demonstrations beforehand, which is not realistic for LfD algorithm deployment. We instead consider the more realistic setting of lifelong learning [37], where an agent adapts to new demos through its lifetime and continually builds its knowledge base. One instance to generate such demonstration sequences is through crowd-sourcing (seeking knowledge from a large set of people) [38, 39, 40].

Despite the abundance of previous approaches, few consider the relationship between the policies learned to represent each demonstration. Our method, FLAIR, exploits these relationships to not only model heterogeneous demonstrations (*adaptability*), but do so by creating expressive policy mixtures from previously extracted strategies (*efficiency*), and can scale to model large number of demonstrations utilizing a compact set of strategies (*scalability*).

## 3   Preliminaries

In this section, we introduce preliminaries on Markov Decision Processes (MDP), Inverse Reinforcement Learning (IRL), and Multi-Strategy Reward Distillation (MSRD).

**Markov Decision Process –** A MDP, $M$, is a 6-tuple, $\langle \mathbb{S}, \mathbb{A}, R, T, \gamma, \rho_0 \rangle$. $\mathbb{S}$ and $\mathbb{A}$ are the state and action space, respectively. $R$ is the reward function, meaning the agent is rewarded $R(s)$ in state $s$. $T(s'|s, a)$ is the probability of transitioning into state $s'$ after taking action $a$ in state $s$. $\gamma \in (0, 1)$ is the temporal discount factor. $\rho_0$ denotes the initial state probability. A policy, $\pi(a|s)$, represents the probability of choosing an action given the state and is trained to maximize the expected cumulative reward, $\pi^* = \arg\max_\pi \mathbb{E}_{\tau \sim \pi} \left[ \sum_{t=1}^{\infty} \gamma^{t-1} R(s_t) \right]$, where $\tau = \{s_1, a_1, s_2, a_2, \cdots\}$ is a trajectory.

**Inverse Reinforcement Learning –** IRL considers an MDP sans reward function (MDP\R) and infers the reward function $R$ based on a set of demonstration trajectories $\mathcal{U} = \{\tau_1, \tau_2, \cdots, \tau_N\}$, where $N$ is the number of demonstrations. Our method is based on Adversarial Inverse Reinforcement Learning (AIRL) [23], which solves the IRL problem with a generative-adversarial setup. The discriminator, $D_\theta$, predicts whether the transition, $(s_t, s_{t+1})$, belongs to a demonstrator vs. the generator, $\pi_\phi(a|s)$. $\pi_\phi$ is trained to maximize the pseudo-reward given by the discriminator.

**Multi-Strategy Reward Distillation –** MSRD [15] assumes access to the strategy label, $c_{\tau_i} \in \{1, 2, \cdots, M\}$ ($M$ is the number of strategies), for each demonstration, $\tau_i$, and decomposes the per-strategy reward, $R_i$, for strategy $i$ as a linear combination of a common task reward, $R_{\text{Task}}$, and a strategy-only reward, $R_{\text{S-}i}$. MSRD parameterizes the task reward by $\theta_{\text{Task}}$ and strategy-only reward by $\theta_{\text{S-}i}$. MSRD takes AIRL as its backbone IRL algorithm, and adds a regularization loss which distills common knowledge into $\theta_{\text{Task}}$ and only keeps personalized information in $\theta_{\text{S-}i}$. The MSRD loss for the discriminator (the reward) is shown in Equation 1.

$$
\begin{aligned}
L_D = &- \mathbb{E}_{(\tau, c_\tau) \sim \mathcal{U}} \left[ \log D_{\theta_{\text{Task}}, \theta_{\text{S-}c_\tau}} (s_t, s_{t+1}) \right] - \mathbb{E}_{(\tau, c_\tau) \sim \pi_\phi} \left[ \log \left( 1 - D_{\theta_{\text{Task}}, \theta_{\text{S-}c_\tau}} (s_t, s_{t+1}) \right) \right] \\
&+ \alpha \mathbb{E}_{(\tau, c_\tau) \sim \pi_\phi} \left[ ||R_{\text{S-}c_\tau}(s_t)||_2 \right]
\end{aligned}
\tag{1}
$$

## 4   Method

In this section, we start by introducing the problem setup and notations. We then provide an overview of FLAIR, and its two key components: *policy mixture* and *between-class discrimination*.

We consider a lifelong learning from heterogeneous demonstration process where demonstrations arrive in sequence, as illustrated in Figure 1. We denote the $i$-th arrived demonstration as $\tau_i$. Unlike

prior work, FLAIR does not assume access to the strategy label, $c_{\tau_i}$. Similar to MSRD, FLAIR learns a shared task reward $R_{\theta_{\text{Task}}}$, strategy rewards $R_{\theta_{\text{S-}j}}$, and policies corresponding to each strategy $\pi_{\phi_j}$. We define the number of prototype strategies created by FLAIR till demonstration $\tau_i$ as $M_i$, and $\eta_R(\tau) = \sum_{t=1}^{\infty} \gamma^{t-1} R_\theta(s_t)$ as trajectory $\tau$'s discounted cumulative reward with the reward function $R_\theta$. The goal of the problem is to accurately model each demonstration sequentially with as few environment samples as possible. Note that learning from sequential demonstrations is not a requirement of FLAIR but rather a feature in comparison to batch-based methods where all demonstrations must be available before the learning could start.

### 4.1 Fast Lifelong Adaptive Inverse Reinforcement Learning (FLAIR)

In our lifelong learning problem setup, when a new demonstration $\tau_i$ becomes available, we seek to accomplish two goals: a) design a policy that solves the task while personalizing to the demonstration (i.e., the objective in personalized LfD), and b) incorporate knowledge from the demonstration to facilitate efficient and scalable adaptation to future users (i.e., the characteristics required for a lifelong LfD framework). We present our method, FLAIR, in pseudocode in Algorithm 1.

---

**Algorithm 1:** FLAIR

**Input** : Demonstration modeling quality threshold $\epsilon$

1   $M_0 = 0$, MixtureWeights=[], $m$=[]
2   **while** *lifetime learning from heterogeneous demonstration* **do**
3      Obtain demonstration $\tau_i$
4      $\vec{w}_i, D_{\text{KL}}^{\text{mix}} \leftarrow \texttt{PolicyMixtureOptimization}(\tau_i, \{\pi_{\phi_j}\}_{j=1}^{M_i})$
5      **if** $D_{KL}^{mix} < \epsilon$ **then**
6         MixtureWeights[i]$\leftarrow \vec{w}_i$, $M_{i+1} \leftarrow M_i$
7      **else**
8         $\pi_{\text{new}}, R_{\theta_{\text{S-}(M_i+1)}} \leftarrow \texttt{AIRL}(\tau_i)$
9         $D_{\text{KL}}^{\text{new}} \leftarrow \mathbb{E}_{\tau \sim \pi_{\text{new}}} D_{\text{KL}}(\tau_i, \tau)$
10        **if** $D_{KL}^{mix} < D_{KL}^{new}$ **then**
11           MixtureWeights[i]$\leftarrow \vec{w}_i$, $M_{i+1} \leftarrow M_i$
12        **else**
13           $M_{i+1} \leftarrow M_i + 1$
14           $m_{M_{i+1}} \leftarrow i$
15           MixtureWeights[i]$\leftarrow [\underbrace{0, 0, \cdots, 0}_{M_i \text{ zeros}}, 1]$
16      Update $R_{\theta_{\text{Task}}}, R_{\theta_{\text{S-}j}}, \pi_{\phi_j}$ by $\texttt{Between-Class Discrimination}$ and $\texttt{MSRD}$

---

To accomplish these goals, FLAIR decides whether to explain a new demonstration with previously learned policies (a highly efficient approach), or create a new strategy from scratch (a fallback technique). In the first case, FLAIR attempts to explain the new demonstration, $\tau_i$, by constructing *policy mixtures* with previously learned strategies according to the demonstration recovery objective (line 4). If the trajectory generated by the mixture is close to the demonstration (evidenced by the KL-divergence between the *policy mixture* trajectory and the demonstration state distributions falling under a threshold, $\epsilon$), FLAIR adopts the mixture without considering creating a new strategy (line 6). Since the *policy mixture* optimization (details in Section 4.2) is more sample efficient than the AIRL training-from-scratch, FLAIR can bypass the computationally expensive new-strategy training (line 8) if the mixture provides a high-quality recovery of the demonstrated behavior. This procedure results in an *efficient* policy inference.

If the mixture does not meet the quality threshold, $\epsilon$, FLAIR trains a new strategy by AIRL with $\tau_i$ and compares the quality of the new policy to the *policy mixture* (Lines 8-10). If the mixture performs better, we accept the mixture weights (line 11). If the new strategy performs better, we accept the new strategy as a new prototype and update our reward and policy models (accordingly, in Line 13, we increment the number of strategies by one). Further, we call the demonstration, $\tau_i$, the "pure" demonstration for strategy $M_{i+1}$, meaning strategy $M_{i+1}$ represents demonstration $\tau_i$ (line 14). As such, the mixture weight for $\tau_i$ is a one-hot vector on strategy $M_{i+1}$ (line 15).

To effectively maintain a knowledge base, we propose a novel training signal named *Between-Class Discrimination* (BCD). BCD trains each strategy reward to capture the fact that each demonstration has a certain percentage of the strategy. In the table tennis example (Figure 1), the third user's behavior is a mixture of the topspin and the slice, indicating topspin and slice strategy rewards should be apparent in the third demonstration. BCD encourages the two strategy rewards to give partial rewards to the third demonstration. In addition to BCD, FLAIR also optimizes MSRD loss (Equation 1) for all strategies with their corresponding pure demonstrations, and updates the generator policies based on the learned reward (line 16).

## 4.2 Policy Mixture Optimization

To achieve efficient personalization for a new demonstration $\tau_i$ (Line 4 of Algorithm 1), we construct a *policy mixture* with a linear geometric combination of existing policies $\pi_1, \pi_2, \cdots, \pi_{M_i}$ (Equation 2), where $w_{i,j} \geq 0$ are learned weights such that: $\sum_{j=1}^{M_i} w_{i,j} = 1$.

$$\pi_{\vec{w}_i}(s) = \sum_{j=1}^{M_i} w_{i,j} a_j, \quad a_j \sim \pi_j(s) \tag{2}$$

As the ultimate goal of demonstration modeling is to recover the demonstrated behavior, we optimize the linear weights, $\vec{w}_i$, to minimize the divergence between the trajectory induced by the mixture policy and the demonstration, shown in Equation 3.

$$\underset{\vec{w}_i}{\text{minimize}} \, \mathbb{E}_{\tau \sim \pi_{\vec{w}_i}} \left[ D_{\text{KL}}(\tau_i, \tau) \right] \tag{3}$$

Specifically, we choose Kullback-Leibler divergence (KL-divergence) [41] on the state marginal distributions of trajectories in our implementation. We estimate the state distribution within a trajectory by the kernel density estimator [42]. More details can be found in supplementary.

Since the trajectory generation process is non-differentiable, we seek a non-gradient-based optimizer to solve Equation 3. Specifically, FLAIR utilizes a naïve, random optimization method; it generates random weight vectors $\vec{w}_i$, evaluates Equation 3, and chooses the weight that achieves the minimization. Empirically, we find random optimization outperforms various other optimization methods for FLAIR. Please see the supplementary for a detailed comparison.

## 4.3 Between-Class Discrimination

Although MSRD distills the task reward from heterogeneous demonstrations, it does not encourage the strategy rewards to encode distinct strategic preferences. MSRD also requires access to ground-truth strategy labels for all demonstrations, which limits scalability. In order to increase the strategy reward's discriminability between different strategies, we propose a novel learning objective named *Between-Class Discrimination* (BCD). BCD enforces the strategy reward to correctly discriminate mixture demonstrations from the pure demonstration: if demonstration $\tau_i$ has weight $w_{i,j}$ on strategy $j$ (as identified in *Policy Mixture*), we could view the probability that $\tau_i$ happens under the strategy reward, $R_{\text{S-}i}$, should be $w_{i,j}$ proportion of the probability of the pure demonstration, $\tau_{m_j}$. This property can be exploited to enforce a structure on the reward given to the pure-demonstration, $\tau_{m_j}$, and mixture-demonstration $\tau_i$, as per Lemma 1. A proof is provided in the supplementary.

**Lemma 1.** *Under the maximum entropy principal,*

$$w_{i,j} = \frac{P(\tau_i; \text{S-}j)}{P(\tau_{m_j}; \text{S-}j)} = \frac{e^{\eta R_{\text{S-}j}(\tau_i)}}{e^{\eta R_{\text{S-}j}(\tau_{m_j})}}$$

Thus, we enforce the relationship of strategy rewards, S-$j$, evaluated on pure strategy demonstration, $\tau_{m_j}$, and mixture strategy demonstration, $\tau_i$ with mixture weight $w_{i,j}$, as shown in Equation 4.

$$L_{\text{BCD}}(\theta^{\text{S-}j}) = \sum_{i=1}^{n} \left( e^{\eta \theta_{\text{S-}j}(\tau_i)} - w_{i,j} e^{\eta \theta_{\text{S-}j}(\tau_{m_j})} \right)^2 \tag{4}$$

An extreme case of BCD loss is when $\tau_i$ is the pure demonstration for another strategy, $k$ (i.e., $m_k = i$). In this case, $w_{i,j} = 0$ (as $\tau_i$ is purely on strategy $k$), and Equation 4 encourages the strategy $j$'s reward to give as low as possible reward to $\tau_i$. In turn, strategy rewards gain better discrimination between different strategies, facilitating more robust strategy reward learning, and contributing to the success in lifelong learning.

Table 1: This table shows learned policy metrics between AIRL, MSRD, and FLAIR. The higher environment returns / lower estimated KL divergence / higher strategy rewards, the better.

| Domains | Inverted Pendulum | | | Lunar Lander | | | Bipedal Walker | | |
|---|---|---|---|---|---|---|---|---|---|
| Methods | AIRL | MSRD | FLAIR | AIRL | MSRD | FLAIR | AIRL | MSRD | FLAIR |
| Environment Returns | $-172.7$ | $-166.4$ | $\mathbf{-38.5}^{**}$ | $-7418.1$ | $-9895.3$ | $\mathbf{-6346.6}^{*}$ | $-30637.2$ | $-74166.0$ | $\mathbf{-7064.0}^{**}$ |
| Estimated KL Divergence | 4.08 | 7.67 | $\mathbf{4.01}^{**}$ | 72.0 | 70.9 | $\mathbf{67.2}^{**}$ | 13.0 | 32.6 | $\mathbf{12.1}^{**}$ |
| Strategy Rewards | $-5.73$ | $-6.22$ | $\mathbf{-1.23}$ | $-12.67$ | $-20.26$ | $\mathbf{-4.19}^{*}$ | $-5.31$ | $-29.82$ | $\mathbf{-4.22}^{**}$ |

$^{*}$ Significance of $p < 0.05$
$^{**}$ Significance of $p < 0.01$

**Correlation between the Estimated and the Ground-Truth Task Reward**

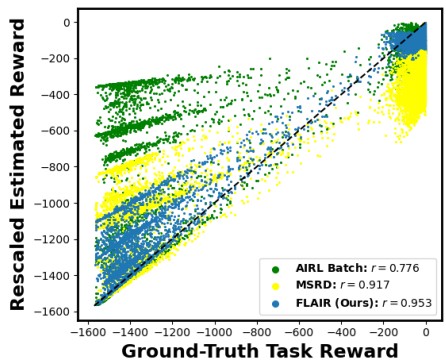

**# Episodes Needed to Achieve the Same Performance**

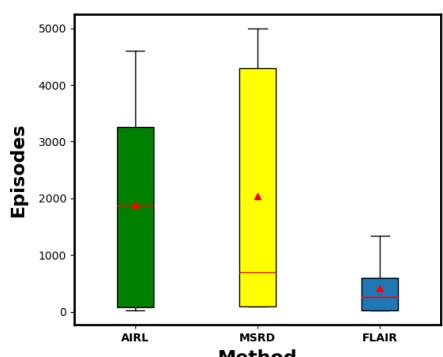

Figure 2: This figure shows the correlation between the estimated task reward with the ground truth task reward for Inverted Pendulum. Each dot is a trajectory. FLAIR achieves a higher task reward correlation.

Figure 3: This figure compares the number of episodes needed for AIRL and MSRD to achieve the same Log Likelihood as FLAIR's mixture optimization. The red bar is the median and the red triangle represents the mean.

## 5 Results

In this section, we show that FLAIR achieves *adaptability*, *efficiency*, and *scalability* in modeling heterogeneous demonstrations. We test FLAIR on three simulated continuous control environments in OpenAI Gym [43]: Inverted Pendulum (IP) [44], Lunar Lander (LL), and Bipedal Walker (BW) [45]. We generate a collection of heterogeneous demonstrations by jointly optimizing an environment and diversity reward with DIAYN [46]. For all experiments excluding the scalability study, we use ten demonstrations. We compare FLAIR with AIRL and MSRD by running three trials of each method. More experiment details and statistical test results are provided in the supplementary.

### 5.1 Adaptability

**Q1:** *Can FLAIR's policy mixtures perform well at the task?* From ten demonstrations, FLAIR created $6.3 \pm 0.5$ strategies (average and standard deviation across three trials) in IP, $5.3 \pm 1.2$ in LL, and $3.3 \pm 0.5$ in BW. FLAIR's learned policies including *policy mixtures* are significantly more successful at the task (row "Environment Returns" in Table 1), outperforming benchmarks in task performance with 77% higher returns in IP, 14% in LL, and 80% in BW than best baselines.

**Q2:** *How closely does the policy recover the strategic preference?* Qualitatively, we find that FLAIR learns policies and policy mixtures that closely resemble their respective strategies, visualized in policy renderings (videos available in supplementary). We further show that FLAIR is statistically significantly better in estimated KL divergence than AIRL (average 4% better) and MSRD (average 18% better), shown in row "Estimated KL Divergence" in Table 1, where KL divergence is evaluated between policy rollouts and demonstration state distributions. We further tested the learned policies' performance on ground-truth strategy reward functions given by DIAYN. The results on row "Strategy Rewards" illustrate FLAIR's better adherence to the demonstrated strategies.

**Q3.** *How well does the task reward model the ground truth environment reward?* We evaluate the learned task reward functions by calculating the correlation between estimated task rewards

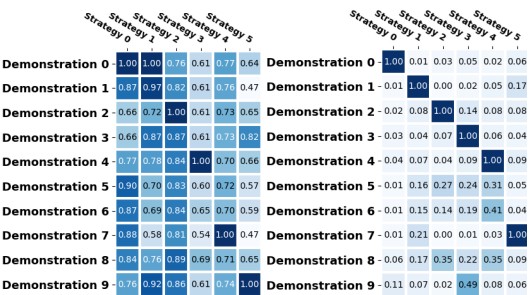
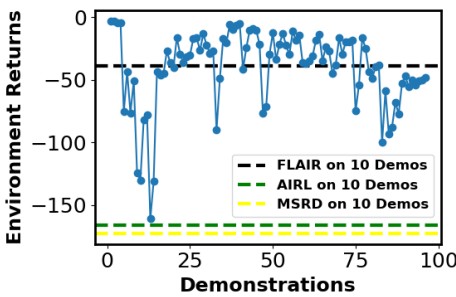

Figure 4: This figure depicts the normalized strategy rewards on demonstrations in IP for FLAIR without BCD (left) and with BCD (right).

Figure 5: This figure plots the returns of FLAIR policies in a 100 demonstration experiment in Inverted Pendulum.

with ground-truth environment rewards. We construct a test dataset of 10,000 trajectories with multiple policies obtained during the "DIAYN+env reward" training. FLAIR's task reward achieves $r = 0.953$ in IP (shown in Figure 2), $r = 0.614$ in LP, and $r = 0.582$ in BW, with an average 18% higher correlation than best baselines and statistical significance compared with AIRL and MSRD.

**Q4. *Can the learned strategy rewards discriminate between different strategies?*** We analyze the learned strategy rewards on heterogeneous demonstrations (shown in Figure 4 right). We find that each strategy reward of FLAIR identifies the corresponding pure demonstration (Demonstrations 0-4,7) alongside the mixtures (Demonstrations 5-6, 8-9). In contrast, the strategy rewards learned without BCD (Figure 4 left) do not distinguish between different strategies. This ablation study also finds that FLAIR with BCD achieves 70% better environment returns and 10% better KL divergence than FLAIR without BCD (additional metrics available in supplementary). The qualitative results in Figure 4 and quantitative results in supplementary together provide empirical evidence that FLAIR with BCD can train strategy rewards to better identify different strategies.

## 5.2 Efficiency & Scalability

**Q5. *Can FLAIR's mixture optimization model demonstrations more efficiently than learning a new policy?*** We study the number of episodes needed by FLAIR's mixture optimization and AIRL/MSRD policy training to achieve the same modeling performance of demonstrations. The result in Figure 3 demonstrates FLAIR requires 77% fewer episodes to achieve a high log likelihood of the demonstration relative to AIRL and 79% fewer episodes than MSRD. Three (out of ten) of AIRL's learned policies and four of MSRD's learned policies failed to reach the same performance as FLAIR even given 10,000 episodes, and are thus left out in Figure 3. By reusing learned policies through *policy mixtures*, FLAIR explains the demonstration in an efficient manner.

**Q6. *Can FLAIR's success continue in a larger-scale LfD problem?*** We generate 95 mixtures with randomized weights from 5 prototypical policies for a total of 100 demonstrations to test how well FLAIR scales. We train FLAIR sequentially on the 100 demonstrations and observe FLAIR learns a concise set of 17 strategies in IP, 10 in LL, and 6 in BW that capture the scope of behaviors while also achieving a consistently strong task performance (Figure 5 and supplementary). We find FLAIR maintains or even exceeds its 10-demonstration performance when scaling up to 100 demonstrations.

## 5.3 Sensitivity Analysis

**Q7. *How sensitive is FLAIR's mixture optimization threshold?*** We study the classification skill of the mixture optimization threshold and find it has a strong ability to classify whether a demonstration should be included as a mixture or a new strategy. A Receiver Operating Characteristic (ROC) Analysis suggests FLAIR with thresholding achieves a high Area Under Curve (0.92) in the ROC Curve for IP; the specific choice of the threshold depends on the performance/efficiency trade-off the user/application demands (see the ROC Curve and threshold selection methodology in the supplementary).

## 5.4 Discussion

The above findings show that our algorithm, FLAIR, sets a new state-of-the-art in personalized LfD. Across several domains, FLAIR achieves better demonstration recovery compared to the baselines.

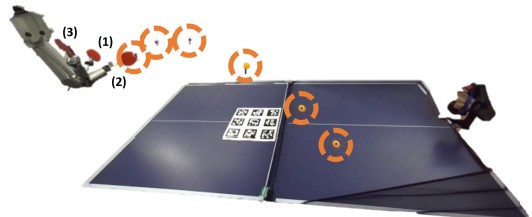

Figure 6: This figure illustrates a topspin and slice mixture policy (a push-like behavior). The robot moves from location (1) to (2) and (3).

| Metrics | Task Score | Strategy Score |
|---|---|---|
| FLAIR's Policy Mixture | $\mathbf{66.9 \pm 10.3}^*$ | $\mathbf{96.6 \pm 17.4}^*$ |
| FLAIR's Worst Mixture | $59.5 \pm 12.8$ | $70.3 \pm 23.7$ |
| Learning-from-Scratch | $56.6 \pm 12.3$ | $90.0 \pm 18.0$ |

$^*$ Significance of $p < 0.05$

Table 2: This table depicts policy metrics between FLAIR's best mixtures, FLAIR's worst mixtures, and learning-from-scratch policies. The scores are shown as averages $\pm$ standard deviations across 28 participants. Bold denotes the highest scores.

Not only can FLAIR more accurately infer the task reward and associated policies, but FLAIR is also able to perform policy inference with much fewer environmental interactions. These characteristics make FLAIR amenable to lifelong LfD, resulting in one of the first LfD frameworks that can handle sequential demonstrations without requiring retraining the entire model.

## 6 Real-World Robot Case Study: Table Tennis

We perform a real-world robot table tennis experiment where we leverage FLAIR's *policy mixtures* to model user demonstrations. An illustration of an example policy mixture is shown in Figure 6 (more videos are available in supplementary).

We first collect demonstrations of four different table tennis strategies (i.e. push, slice, topspin, and lob) via kinesthetic teaching from one human participant who is familiar with the WAM robot but does not have prior experience providing demonstrations for table tennis strikes. After training the four prototypical strategy policies, we assess how well FLAIR can use policy mixtures to model new user demonstrations. To do so, we collected demonstrations from 28 participants by instructing them to demonstrate five repeats of their preferred PingPong strike. We utilize this data and compare three LfD approaches for learning a robot policy: 1) the best policy mixture identified by FLAIR, 2) a learning-from-scratch approach, and 3) an adversarially optimized policy mixture (i.e., minimize the KL divergence between the rollout and the demonstration). We then have users/participants observe the robot executing these policies in a random order. Using ad hoc Likert scale questionnaires (see supplementary), participants evaluate the robot's performance in (i) accomplishing the task and (ii) doing so according to the user's preferences. Table 2 shows that FLAIR's best mixture outperforms both the worst mixture (task score: $p < .01$, strategy score: $p < .001$) and the learning-from-scratch policy (task score: $p < .001$, strategy score: $p < .05$), demonstrating FLAIR's ability to optimize policy mixtures that succeed in the task and fit user's preferences. Full statistical testing results are available in the supplementary.

## 7 Conclusion, Limitations, & Future Work

In this paper, we present FLAIR, a fast lifelong adaptive LfD framework. In benchmarks against AIRL and MSRD, we demonstrate FLAIR's *adaptability* to novel personal preferences and *efficiency* by utilizing policy mixtures. We also illustrate FLAIR's *scalability* in how it learns a concise set of strategies to solve the problem of modeling a large number of demonstrations.

Some limitations of FLAIR are 1) if the initial demonstrations are not representative of a diverse set of strategies, the ability to effectively model a large number of demonstrations may be impacted due to the biased task reward and non-diverse prototypical policies; 2) FLAIR's learned rewards are non-stationary (the learned reward function changes due to the adversarial training paradigm), a property inherited from AIRL, and hence could suffer from catastrophic forgetting. For the first limitation, we could pre-train FLAIR with representative demonstrations before deployment to avoid biasing the task reward and to provide diverse prototypical policies. Another potential direction is to adopt a "smoothing"-based approach over a "filtering" method. The smoothing-based approach would allow new prototypical policies to model previous demonstrations, relaxing the diversity assumptions on initial policies. We are also interested in studying how to recover a minimally spanning strategy set that could explain all demonstrations. For the second limitation, we seek to leverage IRL techniques that yield stationary reward for the FLAIR framework. f-IRL [47] could be a potential candidate, but is notoriously slow due to the iterative reward training and policy training.

**Acknowledgments**

We wish to thank our reviewers for their valuable feedback in revising our manuscript. This work was sponsored by NSF grant IIS-2112633, MIT Lincoln Laboratory grant FA8702-15-D-0001, and Office of Naval Research grant N00014-19-1-2076.

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
