# OpenReview forum: "Fast Lifelong Adaptive Inverse Reinforcement Learning from Demonstrations"
_robot-learning.org/CoRL/2022/Conference — CoRL 2022 Poster_

### Official Review · Reviewer_iYLw · 2022-07-18

**Originality:** Very Good
**Technical Quality:** Very Good
**Clarity Of Presentation:** Good
**Impact:** 3

**Recommendation:**

Weak Accept: I recommend accepting the paper, but will not argue for my recommendation if the majority of other reviewers have a different opinion.

**Summary:**

This paper presents FLAIR, a new algorithm for lifelong “personalized” inverse RL that can rapidly adapt to new heterogeneous demonstrations. They maintain a set of learned policies corresponding to unique skills in demonstrations encountered so far; new demonstrations are modeled as a mixture of existing policies (if the behavior is captured sufficiently well) or a new policy (if the behavior is not). In three simulated continuous control environments, FLAIR outperforms baselines in adaptability, efficiency, and scalability. A real robot experiment is also performed to evaluate the utility of FLAIR's policy mixture.

**Issues:**

1. Attach code supplement if possible
2. Provide Lunar Lander and Bipedal Walker counterparts for all Inverted Pendulum figures
3. Improve Figure 1
4. Explain why real experiment differs from simulation experiment
5. Attach visualizations of heterogeneous demonstrations in simulation if possible

**Quality Of The Limitations Section:**

Limitations are addressed clearly

**Reviewer Expertise:**

3: The reviewer is fairly confident that the evaluation is correct

**Robotics Focus:**

Sufficient demonstration on hardware

**Strengths And Weaknesses:**

This is a strong paper that tackles practical problems in learning from demonstration: lifelong deployment and heterogeneous demonstrations due to varying human preferences. The proposed approach is novel and technically sound, with an intuitive procedure (Algorithm 1) and a novel “Between-Class Discrimination” loss function. Experimental results are organized and presented well, demonstrating a win for FLAIR in adaptability, efficiency, scalability, and policy performance over baseline approaches Adversarial IRL (AIRL) and Multi-Strategy Reward Distillation (MSRD). Experiments suggest FLAIR is a new state-of-the-art approach for lifelong IRL from heterogeneous demonstrations. They also evaluate the quality of FLAIR’s policy mixture in a real robot experiment, and the video supplement gives good visual intuition for the experiment.

While strong, the paper has some weaknesses. Firstly, the authors claim that the code and data will be open-sourced, but it currently is not and is not available in the supplement; it would be ideal to have these available in order to reproduce simulation results, for example. Secondly, the clarity of the writing and presentation can be improved; in particular, Figure 1a is very difficult to parse and the text could benefit from proofreading. Thirdly, the results seem to be weaker for Lunar Lander and Bipedal Walker than the (very simple) Inverted Pendulum environment, as the respective figures are relocated to the appendix (Figure 2 in the appendix shows weak correlation compared to Figure 2 in the main text) or absent from both the text and appendix (e.g., the counterpart for Figure 4). Fourthly, it is unclear why the real robot experiment is a different experiment than the simulations (i.e., the FLAIR vs. AIRL vs. MSRD comparison); some clarification from the authors here would be appreciated.

More minor notes / requests for clarification:
- I am not sure why “crowdsourced” is emphasized in the title and “democratize access to robotics” is emphasized in the Introduction, as there is no large-scale data collection from crowdsourced humans in this paper.
- Have the authors considered a “smoothing” as opposed to “filtering” approach in Algorithm 1, which perhaps could recompute mixture weights for old demos with newer policies that were not available at the time?
- In Section 5.2 under Q6, why does FLAIR recover more than 5 strategies if there are only 5 ground truth policies? Is it approximation error?
- Which environment is the data in Figure 3 from?
- It would be nice to include visualizations of the different heterogeneous demonstrations in simulation for better intuition in Figure 4 and other parts of the paper.


**Summary Of Recommendation:**

I recommend a Weak Accept as this paper makes solid contributions. However, it can be improved by addressing the issues below, and as I am not an expert in personalized IRL I may not be aware of all other existing techniques that may undercut the claim that FLAIR sets a new state-of-the-art.

---

> ### Author Response · Authors · 2022-08-23
> **Response to Reviewer iYLw (Part 1)**
>
> **Comment:**
>
> We thank the reviewer for their helpful feedbacks and suggestions. We are excited that the reviewer thinks our paper is strong in tackling a practical problem, and the experimental results are organized and presented well. Here we address the concerns raised by the reviewer: (edited versions of the paper and the supplementary are uploaded with changes in red)
>
> **Q1: Open-source code**
>
> A1: We wholeheartedly agree reproducibility is a critical facet of research and have thus provided our anonymized code alongside documentation for download at https://tinyurl.com/FLAIRVIDS. We kindly invite the reviewer to inspect our code for any feedback.
>
> **Q2: Improve fig 1 for better clarity**
>
> A2: We have improved Figure 1 in the paper to be more clear and easy-to-follow. We have also made Figure 1 to be on Page 2 on its own without sharing the column space with the previous Figure 1b (now Figure 6).
>
> **Q3.a: Which environment is the data in Figure 3 from?**
>
> A3.a: Figure 3 in the main paper is generated with Inverted Pendulum. We have added counterparts for Lunar Lander and Bipedal Walker for this figure in Supplementary Figure 6 and Figure 7.
>
> **Q3.b: Provide Lunar Lander and Bipedal Walker counterparts for all Inverted Pendulum figures**
>
> A3.b: We thank the reviewer for their suggestion! We have added the counterparts of LL and BW:
>
> 1) for Figure 3 (main paper) as Supplementary Figure 6 and Figure 7
>
> 2) for Figure 4 (main paper) as Supplementary Figure 4 and Figure 5
>
> **Q3.c: Lunar Lander and Bipedal Walker results seem to be weaker than Inverted Pendulum results on task reward correlation**
>
> A3.c: FLAIR’s task reward correlation with ground-truth reward still outperforms baselines by a large margin on Lunar Lander (LL) and Bipedal Walker (BW) in Figure 2 of the supplementary, though FLAIR’s correlation is weaker on LL and BW than on Inverted Pendulum (Figure 2 of the main paper). We hypothesize this may be because recovering the ground-truth shaping reward for LL and BW is harder from heterogeneous demonstrations. FLAIR also outperforms benchmarks on other metrics, including Environment Returns and Estimated KL Divergence, across all three domains in the Table 1 of the main paper.
>
> **Q4: Clarification of why the real robot experiment chooses a different benchmark from simulated experiments**
>
> A4: In the real robot experiment, we tried training AIRL (the simulation experiment benchmark) for each participant with 20 environment episodes, but AIRL fails to produce any meaningful policy on the robot. The very-limited 20-episode budget to adapt to new demonstrations makes AIRL unsuitable for comparing to FLAIR in the real robot experiment. The reason is the same for MSRD as MSRD takes AIRL as the backbone IRL algorithm. Instead, we chose a more meaningful and harder-to-beat learning-from-scratch approach (Supplementary Section 1.3 Lines 36-84) in our real-world table tennis robot, as the proposed learning-from-scratch approach can learn competent strikes in 20 environment episodes.
>
> **Q5: Visualizations of heterogeneous demonstrations in simulation**
>
> A5: We provide videos of heterogeneous demonstrations in simulation in a link in the supplementary page 5 footnote: https://tinyurl.com/FLAIRVIDS for a better visual description.
>
> **Q6: There is no large-scale data collection from crowdsourced humans to claim the title as “learning from crowdsourced demonstrations”**
>
> A6: We thank the reviewer for their feedback. We claimed “crowdsourced demonstrations” as FLAIR theoretically supports the process to personalize to demonstrations obtained from a heterogeneous set of end-users demonstrating tasks asynchronously/sequentially (e.g., through crowdsourcing). Our real-world table tennis experiment illustrates the possibility for FLAIR to model a sequence of 17 human demonstrations on a robot (note we have carried out the experiment with more participants and have updated Table 1 and Section 1.4 in the Supplementary accordingly). We agree such a scale is considerably smaller than a claim for “crowdsourced demonstrations” and have edited the paper accordingly.
>
> --> separating the second part of response due to word limits.
>
> **Zip File:**
>
> /attachment/894a4590bc8bb541c608417543e9ba2e7a1a9426.zip

---

> > ### Author Response · Authors · 2022-08-23
> > **Response to Reviewer iYLw (Part 2)**
> >
> > **Q7: In Section 5.2 Q6, why does FLAIR recover more than five strategies while there are five ground truth policies?**
> >
> > A7: It would require identifying the five ground-truth policies at the beginning of the training and exact optimization for policy mixture for FLAIR to exactly recover the five ground-truth policies. There are multiple approximations in the algorithm: AIRL approximates the ground-truth reward function and the policy[1], neural network approximates the reward/policy, and the Equation 3 approximates the expectation with samples, and therefore it is difficult to recover the exact number of strategies. However, even if the number of strategies is off due to the approximation errors, FLAIR still provides high-quality policies (or policy mixtures) to personalize to each heterogeneous demonstration, as evidenced by our results in simulated experiments and the real-world table tennis experiment.
> >
> > **Q8: “Have the authors considered a “smoothing” as opposed to “filtering” approach in Algorithm 1, which perhaps could recompute mixture weights for old demos with newer policies that were not available at the time?”**
> >
> > A8: We thank the reviewer for their suggestion! Yes, we have considered a smoothing-based approach over a filtering method. The smoothing-based approach would allow later prototypical policies to model previous demonstrations. However, we did not proceed with the smoothing approach due to multiple reasons including: 1) difficulty in handling the demonstrations that were previously modeled with the previous demonstration’s policy; 2) significantly increased computation because of the revisiting process, which limits FLAIR’s ability to learn in lifelong time; 3) previous demonstrations have already been modeled accurate enough by either mixtures or learning-from-scratch. We plan to further investigate the possibility of smoothing in future work. Section 7 Lines 304-305 also mentions that we are interested in studying how to recover a minimally spanning strategy set for all demonstrations as future work.
> >
> > [1] Fu, J., Luo, K., & Levine, S. (2017). Learning robust rewards with adversarial inverse reinforcement learning. arXiv preprint arXiv:1710.11248.

---

> > > ### Comment · Reviewer_iYLw · 2022-08-25
> > > **Thanks**
> > >
> > > Thanks to the authors for their effort in addressing my concerns. I feel that they have been adequately addressed.

---

> > > > ### Author Response · Authors · 2022-08-26
> > > > **Reviewer iYLw Follow-Up**
> > > >
> > > > Thank you for your positive feedback! We are glad to have adequately addressed your concerns. Based upon our revision, we would be grateful if the reviewer might please consider raising the reviewer’s score. Thank you for your consideration.

---

### Official Review · Reviewer_bJrX · 2022-07-24

**Originality:** Good
**Technical Quality:** Good
**Clarity Of Presentation:** Good
**Impact:** 3

**Recommendation:**

Weak Accept: I recommend accepting the paper, but will not argue for my recommendation if the majority of other reviewers have a different opinion.

**Summary:**

This paper proposes an LfD framework to allow adapting to different user preferences over how a task is carried out. It uses an initial set of demonstrations to build an initial set of policies that correspond to different strategies. Then, users specify how they would like the task to be carried out by adding a demonstration. This demonstration is used to infer a policy mixture of the base set of strategies, or learn a new strategy if the demonstration is sufficiently different from the current set of strategies. In this way, the proposed method can learn continually. The proposed method is demonstrated on 3 simulation environments and a real world table tennis environment.

**Issues:**

The main issue is the experimental evaluation - there should be more quantitative (not just qualitative) evidence that the method learns to adhere to new strategies that are demonstrated.

**Quality Of The Limitations Section:**

Additional details required

**Reviewer Expertise:**

4: The reviewer is confident but not absolutely certain that the evaluation is correct

**Robotics Focus:**

Sufficient demonstration on hardware

**Strengths And Weaknesses:**

Strengths

The proposed method is an interesting idea that in principle, allows for more strategies to be learned and modeled as more demonstrations are added into the system. Results on the real-world table tennis domain are impressive (especially those shown in the supplementary video).

Weaknesses

One of the main weaknesses is the experimental evaluation in simulation. The most important aspects of this work, with respect to prior work, appear to be the ability to continually adapt to new strategies (specified by new demonstrations). However, the evaluation of the adherence to strategies is mostly qualitative. It would be very useful to show additional experiments in simulation where there are a collection of ground-truth strategies that are specified by an initial set of humans (similar to the real-world table tennis experiment). Then, quantitative evaluations on how well learned policies adhere to the desired strategy would be possible, instead of the current experiments that rely on unsupervised strategy discovery (through DIAYN, reference 37). From this perspective, it could also be valuable to show results on other domains in simulation where ground-truth strategies can more easily be specified (for example, robotic manipulation, with different grasps possible for objects, or different speeds for trajectories that are executed on the arm).

Related to this, it would be nice to have more quantitative metrics for evaluating adherence to desired strategy in Sec 5.1 (e.g. returns under the ground-truth reward function for the strategy, not just the general task reward function).

More comments follow:

- What if it takes more than one demonstration to show a user-specific strategy? Would the method be able to handle this case?

- Sec 4.2 - how many trajectories are needed to estimate the objective and find the policy mixture weights? Do you then need to collect additional trajectories to estimate the KL divergence for line 5 of algorithm 1, once the mixture weights have been found?

- Sec 6 - how is RL run on the real robot in a sample efficient way? Does the human need to do manual resets? More details on this process would be helpful.

- It's a pretty strong claim to put "crowdsourced demonstrations" in the title without having extensive evaluation with several humans.

- There are a few missing references for crowdsourced demonstrations in the paper (https://arxiv.org/pdf/1811.02790.pdf, https://arxiv.org/abs/1911.04052, https://arxiv.org/abs/2202.02005).

**Summary Of Recommendation:**

The main issue is the experimental evaluation - there should be more quantitative (not just qualitative) evidence that the method learns to adhere to new strategies that are demonstrated. If this issue is addressed, I would increase my score.

**Post Author Response**

Thanks to the authors for addressing my comments in a thorough manner - I have raised my score to Weak Accept.

---

> ### Author Response · Authors · 2022-08-23
> **Response to Reviewer bJrX (Part 1)**
>
> **Comment:**
>
> We thank the reviewer for their constructive feedback and detailed comments. We are excited that the reviewer thinks our work is an interesting idea and that the real-world table tennis experiment is impressive. Here we address the reviewer’s concerns: (edited versions of the paper and the supplementary are uploaded with changes in red)
>
> **Q1: Quantitative metrics for evaluating how FLAIR learns to adhere to desired strategies (e.g., returns under the ground-truth reward function for the strategy)**
>
> A1: To respond to the reviewer’s helpful suggestion, we have additionally tested the new quantitative metric of ***ground-truth strategy reward*** to evaluate the policies learned by FLAIR and benchmarks. Because the demonstrations are generated with the DIAYN strategy reward, we take the strategy reward provided by DIAYN as the ground-truth strategy reward. On the ***ground-truth strategy reward*** metric, FLAIR outperforms both benchmarks on all three simulated domains, calculated across five rollouts on ten strategies. Especially, on Lunar Lander, FLAIR achieves an average strategy reward of -4.19 compared with AIRL (-12.67) and MSRD (-20.26).  We added the “Strategy Rewards” row in Table 1 and Lines 223-225 in the main paper to show FLAIR better learns to adhere to the desired strategies. We also added statistical test results for the Strategy Rewards metric in Table 5 of the supplementary.
>
> We originally utilized an ***Estimated KL Divergence*** metric in Table 1 and Section 5.1 Q2 to serve as our quantitative metric for a policy’s adherence to the corresponding demonstration based upon this metric’s use in prior work [1]. More details about this justification are provided in Supplementary Section 2.1.
>
> We appreciate the suggestion to specify ground-truth strategies by a set of humans and investigate the ability of FLAIR to model such designed strategies. In our paper, we performed the real-world evaluation with real humans and a real, physical robot to demonstrate FLAIR’s personalization capabilities. We show in our real-world table tennis experiment that FLAIR is indeed able to adapt to human task preferences and receive higher task score (16%) and higher strategy score (8%) from participants. In future work, we could also explore simulated domains to compare synthesized and real-human demonstration results.
>
> **Q2: What if people provide more than one demonstration to show a user-specific strategy?**
>
> A2: FLAIR is compatible with one or more demonstrations for a strategy. Specifically, when estimating the KL divergence between the generated trajectory and the demonstration (Equation 3), we pool states in multiple generated trajectories and states from the (one or more) demonstrations. Then, we calculate the estimated KL divergence between the two state-marginal distributions. We have added clarification to Supplementary Lines 141-142. This ability is illustrated in our real-world table tennis experiment, where each participant demonstrates five repeats of their strategies, and FLAIR successfully models the strategies.
>
> **Q3.a: How many trajectories are needed to estimate the policy mixture objective (Equation 3)?**
>
> A3.a: We use three rollouts from the mixture policy to estimate the expectation in Equation 3. The maximum number of the mixture policy weights that FLAIR tries is 900 in simulated experiments and 20 in the real-world table tennis experiment. Supplementary Section 3.4 Lines 216-224 detail hyperparameters for the policy mixture optimization.
>
> **Q3.b: “Do you then need to collect additional trajectories to estimate the KL divergence for line 5 of algorithm 1 once the mixture weights have been found?”**
>
> A3.b: FLAIR does not require additional trajectories to estimate the KL divergence for the best mixture identified, since the KL divergence can be obtained in Pseudocode 1 Line 4. We have updated the pseudocode and the texts accordingly. We thank the reviewer for their feedback!
>
> **Q4: Robot learning details: “how is RL run on the real robot in a sample efficient way? Does the human need to do manual resets?”**
>
> A4: We provide robot learning details in Supplementary Section 1.3 Lines 36-84. RL is too sample inefficient to perform in our experiment with the 20-episode budget to personalize to a user demonstration, as evidenced by the failure described in line 79-81 in the supplementary. Instead, the prototypical policy training is implemented with the approach detailed in Section 1.3.1 in the supplementary. The only manual reset that the system needs is collecting ping-pong balls left on the table after each episode (if the ball becomes still on the table, for example, because of failure to go over-net) to avoid incorrect ball detections in the next episode.
>
> [1] Ni, T., Sikchi, H., Wang, Y., Gupta, T., Lee, L., & Eysenbach, B. (2020). f-irl: Inverse reinforcement learning via state marginal matching. arXiv preprint.
>
> --> separating the second part of response due to word limits.
>
> **Zip File:**
>
> /attachment/7581a2f36c444cedd65d96e9c92982d795bb3de1.zip

---

> > ### Author Response · Authors · 2022-08-23
> > **Response to Reviewer bJrX (Part 2)**
> >
> > **Q5: It is too strong to claim “crowdsourced demonstrations” in the title without having extensive evaluations with several humans.**
> >
> > A5: We thank the reviewer for their suggestions for crowdsourcing-demonstration references. We claimed “crowdsourced demonstrations” as FLAIR theoretically supports the process to personalize to demonstrations obtained from a heterogeneous set of end-users demonstrating tasks asynchronously/sequentially (e.g., through crowdsourcing). Our real-world table tennis experiment illustrates the possibility for FLAIR to model a sequence of 17 human demonstrations on a robot (note we have carried out the experiment with more participants and have updated Table 1 and Section 1.4 in the Supplementary accordingly). We agree such a scale is considerably smaller than a claim for “crowdsourced demonstrations” and have edited the paper accordingly.
> >
> > **Q6: Limitation section**
> >
> > A6: We have added more details to the limitation section (line 295-308, main paper), including clarifications for the non-diverse-initial-demonstration issue and descriptions of the non-stationary issue for AIRL training. We provide a discussion of potential solutions to the limitations, including pretraining, the “smoothing” approach suggested by R3, and leveraging stationary IRL approaches.

---

> > > ### Author Response · Authors · 2022-08-26
> > > **Reviewer bJrX Follow-Up**
> > >
> > > We want to thank reviewer bJrX again for their helpful feedback. Please let us know if we can provide any more information in support of the paper. If our rebuttal has addressed your concerns, we kindly ask the reviewer might consider please increasing the reviewer’s score? Otherwise, we would be happy to continue discussing any remaining items! Thank you again for your review.

---

> > > > ### Comment · Reviewer_bJrX · 2022-08-26
> > > > **Response**
> > > >
> > > > Thanks to the authors for addressing my comments in a thorough manner - I will raise my score to Weak Accept.

---

> > > > > ### Author Response · Authors · 2022-08-27
> > > > > **Thank You**
> > > > >
> > > > > We are glad you find our response thoroughly addressed your concerns. Thank you so much again for taking the time to review our paper as well as our responses!

---

### Official Review · Reviewer_rqFS · 2022-08-01

**Originality:** Good
**Technical Quality:** Very Good
**Clarity Of Presentation:** Very Good
**Impact:** 3

**Recommendation:**

Weak Accept: I recommend accepting the paper, but will not argue for my recommendation if the majority of other reviewers have a different opinion.

**Summary:**

The paper proposes Fast Lifelong Adaptive IRL (FLAIR), a learning from demonstration framework that aims to maintain a collection of learned strategies that may be mixed in order to model subsequent demonstrations. The method is similar to Multi-Strategy Reward Distillation (MSRD) but does not assume access to a strategy label. Overall, the paper is well-written and provides a thorough experimental evaluation.

**Issues:**

Please see weaknesses/questions listed above and address accordingly.

**Quality Of The Limitations Section:**

Additional details required

**Reviewer Expertise:**

3: The reviewer is fairly confident that the evaluation is correct

**Robotics Focus:**

Sufficient demonstration on hardware

**Strengths And Weaknesses:**

Strengths:
- The paper proposes a new method for constructing a policy mixture for a new demonstration as well as a new objective, Between-Class Discrimination (BCD) that seems to be significantly more effective than existing work at adapting to a sequence of demonstrations.
- The paper conducts a thorough experimental evaluation, which includes ablations on FLAIR and even a real robot experiment with a table tennis-playing robot. The paper also includes a very thorough appendix.

Weaknesses:
- It seems like the effectiveness of the method may depend on the range of demonstrations available--if they are not diverse, then a lack of strategies is modeled, and if they are too diverse, the method may learn too many strategies.
- I'm not sure about the importance of the problem setting, and I think the assumptions made in the problem setting could be motivated better. It's not clear to me how many real world scenarios are actually modeled by this problem statement where individual demonstrations must arrive in sequence. For example, a robot learning to play table tennis will likely have access to a library of demonstrations to start instead of requiring each demonstration to arrive in sequence.

Questions:
- As a baseline, what is the performance when not given demos in a sequential manner, e.g. given demos all at once, with AIRL or GAIL?
- It would be helpful to add titles to the plots in Figure 2 and 3.


**Summary Of Recommendation:**

Overall, despite some weaknesses, I think this is an interesting paper with a novel method and well-structured experiments.

---

> ### Author Response · Authors · 2022-08-23
> **Response to Reviewer rqFS**
>
> **Comment:**
>
> We thank the reviewer for their thoughtful feedback. We are excited that the reviewer found our work well-written with a thorough evaluation. We would like to address the reviewer’s concerns below: (edited versions of the paper and the supplementary are uploaded with changes in red)
>
> **Q1.a: Importance of the problem setting for learning from sequential demonstrations**
>
> A1.a: One real-world example of our problem setup is household robots delivered to users’ homes. Users will want to teach these robots new skills over the course of the deployment.  User demonstrations from different end-users during the process form a demonstration sequence, and the robot will need to personalize to new demonstrations given by specific end-users (adaptation), utilize prior learned information to quickly learn robot behaviors preferred by specific end-user (efficiency), and be able to represent learned knowledge effectively for later users that receive the household robot (scalability)[1]. We added this example in lines 53-56 to the paper for better motivation of the setup.
>
> **Q1.b: "It is not clear to me how many real-world scenarios are actually modeled by this problem statement where individual demonstrations must arrive in sequence."**
>
> A1.b: Our framework, FLAIR, can utilize an initial library of demonstrations to bootstrap the learning process, as shown in our real-world table tennis experiment. However, FLAIR can also start from scratch with a single demonstration and grow one-by-one. Handling sequential demonstrations is not a requirement of FLAIR but rather is a feature in comparison to batch-based methods. We added this clarification in lines 131-134 to the main paper.
>
> **Q2: The effectiveness of FLAIR may depend on the range of demonstrations available.**
>
> A2: We agree. We state in the limitation section lines 293-295 that “if the initial demonstrations are not representative diverse strategies, the learned task reward might be biased, and the ability to effectively model a large number of demonstrations (expressivity of the policy mixtures) might be impacted.” However, in the household robot example in A1.a (main paper lines 53-56), if we assume the users are drawn independently from the same distribution (i.i.d), FLAIR can adapt to a user set and perform well on the user preference distribution. For example, if the set of users performs diverse demonstrations, FLAIR can create more strategies to model them and will perform well with future users with diverse demonstrations. Instead, if users demonstrate strategies that are well-modelled by a combination of prototypes learned early in the process, FLAIR will create fewer strategies. Thus, FLAIR creates strategies when needed, and if it creates “too many” strategies, it may indicate that the demonstrations are diverse rather than there necessarily being “too many” strategies.
>
> **Q3. Benchmark of giving demonstrations all at once**
>
> A3: We would like to kindly mention that, in Supplementary Section 3.2 line 191-193, we describe our experiments noting that demonstrations are given all at once for AIRL: “On policy evaluation metrics, we train AIRL on each individual demonstration (named AIRL Single) in favor of its personalization. On reward metrics, we train AIRL on the entire set of demonstrations (named AIRL Batch) to improve its reward robustness.”
>
> In the main paper, we show in Table 1 that **FLAIR outperforms AIRL Single on personalization metrics** (Estimated KL Divergence and Strategy Rewards), and AIRL Single should perform better than AIRL Batch for personalization as AIRL Batch only trains one policy for all demonstrations. In Figure 2, we show that **FLAIR outperforms AIRL Batch on task reward metric** (Correlation between the Estimated and the Ground-Truth Task Reward), and AIRL Batch should perform better than AIRL Single for task reward learning as AIRL Batch’s reward function has access to all demonstration information.
>
> Thus, we conclude FLAIR outperforms AIRL when giving demonstrations all at once.
>
> **Q4. Add titles to plots in Fig 2 and 3 for clarity**
>
> A4: We added titles to Figures 2 and 3. We thank the reviewer for their suggestion!
>
> **Q5. Additional details of the limitations are quired**
>
> A5: We have added more details to the limitation section (line 295-308, main paper), including clarifications for the non-diverse-initial-demonstration issue and descriptions of the non-stationary issue for AIRL training. We provide a discussion of potential solutions to the limitations, including pretraining, the “smoothing” approach suggested by R3, and leveraging stationary IRL approaches.
>
>
> [1] Charlie Tritschler, V. P. of P. at A. (2021, September 28). Meet Astro, a home robot unlike any other. US About Amazon. Retrieved August 19, 2022, from https://www.aboutamazon.com/news/devices/meet-astro-a-home-robot-unlike-any-other
>
> **Zip File:**
>
> /attachment/2193365ee64011e52ebe00fe84de0bc669003429.zip

---

> > ### Author Response · Authors · 2022-08-26
> > **Reviewer rqFS Follow-Up**
> >
> > We want to thank reviewer rqFS again for their helpful feedback. Please let us know if we can provide any more information in support of the paper. If our rebuttal has addressed your concerns, we kindly ask the reviewer might consider please increasing the reviewer’s score? Otherwise, we would be happy to continue discussing any remaining items! Thank you again for your review.

---

### Meta-Review · Area_Chair_myNd · 2022-08-14

**Recommendation:** Accept (Poster)
**Confidence:** 4

**Metareview:**

Phase 1:

Strengths:
The submission provides a new and technically relevant method. It is well structured and intuitive in its argument. It provides a thorough experimental evaluation including real world robot experiments in a challenging domain.

Weaknesses:
Some weaknesses are found in the evaluation and multiple reviewers ask about clarifications. In particular, these include the different choices for comparisons and baselines between simulation and real experiments, weaker results on some of the toy domains, as well as quantitative metrics for adapting to different strategies.

Phase 2:

The feedback has originally been a borderline case with slightly more positive than negative feedback. Many points could be addressed during the review and final reviews are generally positive (3 weak accepts). The reviews point out that it is well structured and intuitive in its argument. It provides a thorough experimental evaluation including real world robot experiments in a challenging domain but also required clarifications around the evaluations and confusion about some of the baselines. I agree with the reviewers and recommend acceptance. Please take the remaining points from the review process seriously and follow up with improvements on open points and promised changes.

**Best Paper Nomination:**

No

---

> ### Author Response · Authors · 2022-08-23
> **Response to Meta Reviewer Area Chair myNd**
>
> **Comment:**
>
> We thank the meta-reviewer for taking the time to review our work. We are working on providing clarifications for the reviewers’ questions in detail. In particular, we would like to highlight the major explanations here:
>
> **Q1: Clarification of why the real robot experiment chooses a different baseline from simulated experiments**
>
> A1: We chose AIRL and MSRD to benchmark in the simulation experiments and chose a learning-from-scratch method (detailed in Section 1.3, Supplementary) to benchmark in the real-world experiment. The reason why the real-world experiment has a different benchmark is that AIRL and MSRD do not produce any meaningful policy for the table tennis in the real-world experiment setup, which is to personalize to each user demonstration with a budget of twenty environment interaction episodes. If we were to benchmark against AIRL/MSRD, we expected AIRL/MSRD would receive near-minimum ratings from the participants. The learning-from-scratch benchmark we introduced could instead provide meaningful table tennis strikes with the experiment setup. Thus, the learning-from-scratch approach is a harder-to-beat and more meaningful baseline in the real-world table tennis experiment.
>
> **Q2: Lunar Lander and Bipedal Walker results seem to be weaker than Inverted Pendulum results on task reward correlation**
>
> A2: FLAIR’s task reward correlation with ground-truth reward still outperforms baselines by a large margin on Lunar Lander (LL) and Bipedal Walker (BW) in Figure 2 of the supplementary, though FLAIR’s correlation is weaker on LL and BW than on Inverted Pendulum (Figure 2 of the main paper). We hypothesize this may be because recovering the ground-truth shaping reward for LL and BW is harder from heterogeneous demonstrations. FLAIR also outperforms benchmarks on other metrics, including Environment Returns and Estimated KL Divergence, across all three domains in the Table 1 of the main paper. We have provided additional result plots per R2’s and R3’s suggestions.
>
> **Q3: Quantitative metrics for adapting to strategies**
>
> A3: To respond to the reviewer’s helpful suggestion, we have additionally tested the new quantitative metric of ***ground-truth strategy reward*** to evaluate the policies learned by FLAIR and benchmarks. Because the demonstrations are generated with the DIAYN strategy reward, we take the strategy reward provided by DIAYN as the ground-truth strategy reward. On the ***ground-truth strategy reward*** metric, FLAIR outperforms both benchmarks on all three simulated domains, calculated across five rollouts on ten strategies. Especially, on Lunar Lander, FLAIR achieves an average strategy reward of -4.19 compared with AIRL (-12.67) and MSRD (-20.26).  We added the “Strategy Rewards” row in Table 1 and Lines 223-225 in the main paper to show FLAIR better learns to adhere to the desired strategies. We also added statistical test results for the Strategy Rewards metric in Table 5 of the supplementary.
>
> We originally utilized an ***Estimated KL Divergence*** metric in Table 1 and Section 5.1 Q2 to serve as our quantitative metric for a policy’s adherence to the corresponding demonstration based upon this metric’s use in prior work. More details about this justification are provided in Supplementary Section 2.1.
>
> Might the meta-reviewer please be able to share with us if more quantitative metrics beyond what is shown would be helpful?
>
>
> **Zip File:**
>
> /attachment/2ec4a8d61f19e39cb86e29570da982a821235149.zip